# From Metabolic Syndrome to Cardio-Kidney-Metabolic Syndrome in the SIMETAP Study: Prevalence Rates of Metabolic Syndrome and Its Independent Associations with Cardio-Renal-Metabolic Disorders Other than Its Defining Criteria

**DOI:** 10.3390/biomedicines13030590

**Published:** 2025-02-28

**Authors:** Antonio Ruiz-García, Ezequiel Arranz-Martínez, Adalberto Serrano-Cumplido, Sergio Cinza-Sanjurjo, Carlos Escobar-Cervantes, José Polo-García, Vicente Pallarés-Carratalá

**Affiliations:** 1Lipids and Cardiovascular Prevention Unit, Pinto University Health Centre, 28320 Madrid, Spain; 2Department of Medicine, European University of Madrid, 28005 Madrid, Spain; 3San Blas Health Centre, 28981 Madrid, Spain; ezequielarranz@gmail.com; 4Repelega Health Centre, 48920 Portugalete, Spain; adal1953@hotmail.com; 5Milladoiro Health Centre, Health Area of Santiago de Compostela, 15895 Ames, Spain; scinzasanjurjo@gmail.com; 6Department of Medicine, University of Santiago de Compostela, 15706 Santiago de Compostela, Spain; 7Department of Cardiology, La Paz University Hospital, 28046 Madrid, Spain; carlos.escobar@salud.madrid.org; 8Casar de Cáceres Health Centre, 10190 Cáceres, Spain; jpolog@semergen.es; 9Department of Medicine, Jaume I University, 12006 Castelló de la Plana, Spain

**Keywords:** adults, arterial hypertension, cardiovascular-kidney-metabolic syndrome, chronic kidney disease, diabetes mellitus, excess adiposity, metabolic syndrome, prevalence

## Abstract

**Background/Objectives**: Metabolic syndrome (MetS) is a highly prevalent entity defined according to cardiometabolic criteria. Other disorders related to MetS could help assess the comprehensive risk of diabetes, cardiovascular disease, and chronic kidney disease (CKD). This study aimed to update the prevalence rates of MetS and to assess its relationship with other disorders and clinical conditions other than the criteria defining MetS. **Methods**: A cross-sectional observational study was conducted with a random population-based sample of 6588 study subjects between 18 and 102 years of age. Crude and sex- and age-adjusted prevalence rates of MetS were calculated, and their associations with comorbidities and clinical conditions other than their defining criteria were assessed by bivariate and multivariate analysis. **Results**: The adjusted prevalence rates were 36.0% for MetS (39.8% in men; 33.5% in women), 21.5% for premorbid Mets, and 14.5% for morbid MetS. Considering only clinical conditions other than the criteria defining MetS, the independent disorders associated with premorbid MetS were hypercholesterolemia, hypertension, high levels of lipid accumulation product, high triglyceride-glucose index (TyG), high visceral adiposity index, high fatty liver index, and high waist-to-height ratio (WtHR), highlighting excess adiposity (EA). The independent disorders associated with morbid MetS were hypercholesterolaemia, high-WtHR, EA, high-TyG index, heart failure, atrial fibrillation, CKD, and albuminuria, highlighting hypertension. **Conclusions**: One-fifth of the adult population has premorbid MetS, and almost one-sixth has morbid MetS. Almost two-fifths of people with MetS are at moderate, high, or very high risk of CKD, and four-fifths are at high or very high cardiovascular risk. In addition to the criteria defining MetS, other cardiovascular-renal-metabolic disorders show an independent association with MetS, highlighting EA for premorbid MetS and hypertension for morbid MetS.

## 1. Introduction

Metabolic syndrome (MetS) is a complex clinical entity of interconnected cardiometabolic disorders that includes abdominal obesity, high blood pressure, atherogenic dyslipidaemia, and hyperglycaemia. MetS is a major health problem due to its high prevalence [1,2] and because it increases the risk of developing diabetes (DM) five-fold and the risk of atherosclerotic cardiovascular disease (ASCVD) almost three-fold [3,4].

In the 1920s, the Swede Eskil Kylin [5] and the Spaniard Gregorio Marañón [6] published studies on prediabetic states in which arterial hypertension (HTN), obesity, hyperuricaemia, and hyperglycaemia coexisted. In 1967, Avogaro et al. [7] showed the association between hyperlipidaemia, DM, and obesity. In 1988, Gerald Reaven [8] highlighted the importance of the cardiovascular risk of an entity called “syndrome X”, whose pathogenic core was insulin resistance and which was characterised by a set of metabolic disorders that included hyperglycaemia, HTN, increased plasma triglyceride levels, and decreased plasma concentration of high-density lipoprotein cholesterol (HDL-c). Finally, the World Health Organization (WHO) [9] named this entity as MetS. In 2001, the Expert Panel on Detection, Evaluation, and Treatment of High Blood Cholesterol in Adults (Adult Treatment Expert Panel III) issued the Third Report of the National Cholesterol Education Program (NCEP/ATP-III), which defined five criteria for MetS [10]. In 2004, the American Heart Association and the National Heart, Lung, and Blood Institute (AHA/NHLBI) [11] lowered the fasting plasma glucose (FPG) criterion from ≥ 110 mg/dL to ≥ 100 mg/dL. In 2005, the International Diabetes Federation (IDF) maintained the five criteria for MetS except for waist circumference (WC) [12]. The 2009 Joint Statement of the IDF, AHA/NHLBI, World Heart Federation, International Atherosclerosis Society, and International Association for the Study of Obesity (2009-JS) updated the new criteria for MetS, which they have adopted as definitive, allowing them to be unified and thus to be able to compare studies worldwide, highlighting that those cut-off points for high WC should be adapted to different ethnic considerations of the populations [13].

Despite this consensus, disagreements regarding how to understand MetS continued. While NCEP/ATP-III [10] stated that the presence of type 2 DM did not exclude the diagnosis of MetS, WHO experts considered that MetS should be considered a premorbid condition and, therefore, individuals with DM or ASCVD should be excluded from the MetS diagnosis [14]. Cardiovascular-kidney-metabolic (CKM) syndrome [15] is a complex entity of interactions among the cardiovascular, renal, and metabolic systems, which assesses the comprehensive risk of DM, ASCVD, and CKD better than MetS. Other disorders included in the CKM syndrome such as obesity, excess adiposity (EA), HTN, heart failure (HF), atrial fibrillation (AF), albuminuria, or chronic kidney disease (CKD) could be related to MetS. Our study aimed to update the prevalence rates of MetS and to assess its relationship with other disorders and clinical conditions other than the criteria defining MetS.

## 2. Materials and Methods

### 2.1. Study Design

We conducted a cross-sectional observational multicentre sub-study from the SIMETAP study, whose design and methodology were previously published [16] and authorised by the Madrid Health Service (SERMAS, according to its acronym in Spanish). The Healthcare Identity Cards (HIC) census assigned to SERMAS was 99% of the population (5,144,860 adults) of the Region of Madrid (Spain), whose healthcare was provided in 260 primary healthcare centres. Briefly, a simple random sampling was performed using Excel’s randbetween function on HIC assigned to the research physicians (194,073 adults) until reaching the sample size necessary to evaluate the study aims (response rate 62.9%). One hundred and twenty-one physicians from 64 healthcare centres collected data and conducted interviews with participants. Inclusion criteria: adults (≥18 years of age) with informed consent and the clinical and laboratory data necessary to be assessed. Patients with terminal illnesses or cognitive impairment, dementia, schizophrenia, moderate or severe psychosis, residents in nursing homes, pregnant women, and people who are participating in other clinical studies were excluded according to the protocol approved by the Research Ethics Committee (Code 05/2010RS). The flowchart for sampling and selection of study subjects is shown in Figure 1. All information assessed in the study was collected from the primary care electronic health records in a real-world data setting.

### 2.2. Assessment Variables

The primary endpoint was MetS diagnosis according to 2009-JS for the European population [13], which requires the presence of three or more of the following criteria: WC ≥102 cm for men or ≥88 cm for women; triglycerides ≥150 mg/dL (≥1.7 mmol/L); HDL-c < 40 mg/dL (<1.03 mmol/L) for men or <50 mg/dL (<1.29 mmol/L) for women; systolic blood pressure (SBP) ≥ 130 mmHg or diastolic blood pressure (DBP) ≥ 85 mmHg (blood pressure-lowering drug therapy [BPLT] in a patient with a history of HTN was an alternate indicator); FPG ≥ 100 mg/dL (≥5.6 mmol/L) (glycaemic-lowering drug therapy [GLT] for elevated glucose was an alternate indicator). The secondary endpoints were premorbid MetS (absence of DM or ASCVD) and morbid MetS (presence of DM and/or ASCVD). The criteria and definitions of the diseases or clinical conditions assessed are reported in detail in Appendix A (Appendix A).

### 2.3. Statistical Analysis

Prevalence rates were determined for the overall adult study population and according to age-groups. The age- and sex-adjusted prevalence rates were calculated by the direct method, according to Spanish population data from the National Institute of Statistics of Spain. The frequency and percentage of qualitative variables and the mean and standard deviation (SD) of quantitative variables were reported. Percentages and odds ratios (OR) were used with a 95% confidence interval (CI). Comparisons of percentages were performed using the Chi-square test or Fisher’s exact test when at least 20% of the expected frequencies were less than five. A Shapiro–Wilk test was used to check the data fitting to normal distribution for continuous variables. If the variables showed normal distribution, they were compared using Student’s *t*-test or analysis of variance. Cohen’s *d* was used to assess the effect size of standardised mean differences, according to the proximity to the following absolute *d*-values: 0.2 small; 0.5 medium; and 0.8 large. Multivariate logistic regression analyses were performed using the backward stepwise method to assess the individual effect of comorbidities and clinical conditions on the dependent variables (MetS, premorbid MetS, and morbid MetS). All variables that showed an association in the bivariate analysis up to a *p*-value < 0.10 were included in the model, except those that could bias it, such as the variables of MetS criteria, and erectile dysfunction because it only affects men. Subsequently, the variable that contributed the least to the fit of the analysis was eliminated at each step. All analyses were performed with SPSS Statistics for Windows, version 25 (IBM Corporation, Armonk, NY, USA). All statistical tests were two-tailed with a *p*-value < 0.05 used to determine statistical significance.

## 3. Results

### 3.1. Study Population

Our study included 6588 people (55.9% women) ranged 18.0 to 102.8 years of age. Their mean (SD) age was 55.1 (17.5) years, with a non-significant difference between men (55.3 [16.9] years) and women (55.0 [18.0] years) (*p* = 0.634). Among study subjects, 15.7% (95% CI 14.8–16.6) had DM, 9.3% (95% CI 8.6–10.0) had ASCVD, 21.4% (95% CI 20.4–22.4) had DM or ASCVD, and 11.5% (95% CI 10.7–12.3) had CKD. Female percentages (95% CI) among people with MetS, premorbid MetS, and morbid MetS were 51.2% (49.3–53.0), 55.6% (53.2–58.1), and 45.3% (42.4–48.1), respectively. Comparisons of percentages and ORs of MetS criteria [13] between populations with and without MetS are shown in Table 1.

### 3.2. Prevalence Rates of MetS

Crude and adjusted prevalence rates are shown in Table 2. Prevalence rates were significantly higher in men than in women with MetS according to the 2009-JS (*p* < 0.001), NCEP/ATP-III (*p* < 0.001), and IDF (*p* = 0.001) criteria [10,12,13]. Prevalence rates were significantly higher in men than in women with morbid MetS (*p* < 0.001), according to the 2009-JS criteria, and were similar for people with premorbid MetS (*p* = 0.777). The age-groups distributions of prevalence rates for MetS, premorbid MetS, and morbid MetS according to the 2009-JS criteria [13] (Figure 2a–c), and the distributions according to the NCEP/ATP-III [10] and IDF criteria [12] (Appendix A, Appendix A) increased precisely with age (R^2^ > 0.99) according to polynomial functions. Age-specific prevalence rates for MetS according to the 2009-JS criteria were significantly higher in men for all age-groups up to 59 years of age (Figure 2a), up to 49 years of age for premorbid MetS (Figure 2b), and up to 79 years of age for morbid MetS (Figure 2c). Age-specific prevalence rates were significantly higher in women aged 60 years and older for premorbid MetS (Figure 2b). Likewise, the age-groups distributions of prevalence rates for MetS according to the NCEP/ATP-III [10] and IDF criteria [12] were significantly higher (*p* < 0.001) in men for all age-groups up to 59 years of age (Appendix A, Appendix A).

### 3.3. Analysis for Populations with and Without MetS

Differences in mean age between both populations was 16.5 yr (95% CI 15.8–17.3). All clinical variables were significantly higher (*p* < 0.001) in the population with MetS than in the non-MetS population, except HDL-c, low-density lipoprotein cholesterol (LDL-c), and estimated glomerular filtration rate (eGFR), which were significantly higher (*p* < 0.001) in the non-MetS population. Differences in total cholesterol (TC) and aspartate aminotransferase (AST) were non-significant between both populations. Standardised mean difference effect sizes were large for body mass index (BMI), WC, waist-to-height ratio (WtHR), CUN-BAE (according to its acronym in Spanish, *Clínica Universitaria de Navarra*—Body Adiposity Estimator) EA, SBP, FPG, glycated haemoglobin A1c (HbA1c), triglycerides, atherogenic index of plasma (AIP), triglyceride-glucose (TyG) index, lipid accumulation product (LAP) index, visceral adiposity index (VAI), fatty liver index (FLI), and eGFR; and were medium or small for the remaining diseases and clinical conditions (Table 3). Bivariate analysis for comorbidities and clinical conditions in populations with and without MetS showed ORs between 2.9 and 4.4 for erectile dysfunction, hyperuricaemia, albuminuria, low eGFR, and CKD; ORs between 5.2 and 9.3 for obesity, high-WtHR, high-VAI, hypercholesterolaemia, high-AIP, high-LAP index, high-TyG index, high-FLI, stroke, peripheral arterial disease (PAD), ASCVD (includes coronary heart disease [CHD], stroke and PAD), HF, AF, and cardiovascular diseases (CVD) (includes ASCVD, HF and AF); and highlighted HTN, CUN-BAE EA, DM, and CHD, with ORs 11.7, 16.0, 16.2, and 17.2, respectively (Figure 3a, Appendix A, Appendix A). Among subjects with MetS, 32.6% (95% CI 30.8–34.3) had DM, 18.4% (95% CI 17.0–19.9) had ASCVD, 42.9% (95% CI 41.0–44.7) had DM or ASCVD, and 19.6% (95% CI 18.6–21.1) had CKD; 27.0% (95% CI 25.4–28.7%) were on GLT, 60.5% (95% CI 58.7–62.3) were on lipid-lowering drug therapy (LLT), 64.0% (95% CI 62.3–65.8) were on BPLT, and 3.7% (95% CI 3.1–6.8) were on urate-lowering drug therapy (ULT) (Appendix A, Appendix A).

### 3.4. Analysis for Populations with Morbid MetS vs. with Premorbid MetS

Differences in mean age between both populations was 7.7 yr (95% CI 6.7–8.7). Differences in WC, WtHR, FGP, HbA1c, TyG index, creatinine, and urine albumin–creatinine ratio (uACR) were significantly higher in the population with morbid MetS than in the population with premorbid MetS. DBP, TC, LDL-c, and eGFR were significantly higher (*p* < 0.001) in the population with premorbid MetS. Standardised mean difference effect sizes were large for FPG, HbA1c, TC, and LDL-c and were medium or small for the remaining variables (Table 4). Percentages for high-WtHR, hypercholesterolaemia, high-TyG index, HF, AF, CVD, erectile dysfunction, albuminuria, and low eGFR and CKD were significantly higher (*p* < 0.001) in the population with morbid MetS than in the population with premorbid MetS, and on the contrary, were significantly lower (*p* < 0.001) for HTN and prediabetes (Appendix A, Appendix A).

### 3.5. Analysis for Populations with Morbid MetS vs. Non-MetS

Differences in mean age between both populations was 20.9 yr (95% CI 19.9–21.9). Differences in most clinical variables were significantly higher (*p* < 0.001) in the population with morbid MetS than in the non-MetS population, except TC, HDL-c, LDL-c, and eGFR, which were significantly higher (*p* < 0.001) in the non-MetS population. Standardised mean difference effect sizes were large for most clinical variables and medium or small for DBP, TC, HDL-c, LDL-c, TG/HDL-c, SUA, GGT, creatinine and uACR (Table 4). Bivariate analysis for comorbidities and clinical conditions in the population with morbid MetS vs. non-MetS showed OR 3.0 for hyperuricaemia; ORs between 5.5 and 9.6 for obesity, high-WtHR, hypercholesterolaemia, high-AIP, high-LAP, high-TyG index, high-FLI, AF, erectile dysfunction, albuminuria, low eGFR, and CKD; and highlighted HF and CUN-BAE EA, with ORs 11.0 and 17.0, respectively (Figure 3b, Appendix A, Appendix A). Among patients with morbid MetS, 75.9% (95% CI 73.5–78.3) had DM, 43.0% (95% CI 40.3–45.8) had ASCVD, and 19.0% (95% CI 16.8–21.2) had DM and ASCVD; 62.6% (95% CI 59.9–65.3%) were on GLT, 76.4% (95% CI 74.0–78.7) were on LLT, 77.7% (95% CI 75.4–80.1) were on BPLT, and 4.2% (95% CI 3.0–5.3) were on ULT (Appendix A, Appendix A).

### 3.6. Analysis for Populations with Premorbid MetS vs. Non-MetS

Differences in mean age between both populations was 13.2 yr (95% CI 12.3–14.1). Differences in all clinical variables were significantly higher (*p* < 0.001) in the population with premorbid MetS than in the non-MetS population, except HDL-c and eGFR, which were significantly higher (*p* < 0.001) in the non-MetS population. Differences for AST were non-significant (*p* = 0.195). Standardised mean difference effect sizes were large for BMI, WC, WtHR, CUN-BAE, SBP, AIP, TyG index, LAP, VAI, and FLI; and medium or small for the remaining variables assessed (Table 4). Bivariate analysis for comorbidities and clinical conditions in the population with premorbid MetS vs. non-MetS showed ORs between 1.8 and 3.3 for HTN, HF, AF, erectile dysfunction, hyperuricaemia, albuminuria, low eGFR, and CKD; between 4.8 and 10.0 for obesity, high-WtHR, high-VAI, hypercholesterolaemia, high-AIP, high-LAP, high-TyG index, and high-FLI; and highlighting CUN-BAE EA with OR 15.3 (Figure 3c, Appendix A, Appendix A). Among patients with premorbid MetS, 0.4% (95% CI 0.1–0.7%) were on GLT, 48.7% (95% CI 46.3–51.1) were on LLT, 53.8% (95% CI 51.3–56.2) were on BPLT, and 3.4% (95% CI 2.5–4.3) were on ULT (Appendix A, Appendix A).

### 3.7. Independent Associations with Disorders Other than the Criteria Defining MetS

Multivariate analysis showed that HTN, hypercholesterolemia, CUN-BAE AE, and high-WtHR were independently associated with MetS and with both morbid and premorbid MetS patterns. In addition, obesity, high-VAI, high-TyG index, and high-FLI were also independently associated with MetS, highlighting DM (Figure 4a, Appendix A, Appendix A); high-TyG index, HF, AF, albuminuria, and CKD were independently associated with morbid MetS, highlighting HTN (Figure 4b, Appendix A, Appendix A); and high-VAI, high-LAP, and high-FLI were also independently associated with premorbid MetS, highlighting CUN-BAE AE (Figure 4c, Appendix A, Appendix A).

### 3.8. Analysis of Risks for CKD and for CVD

The percentages of subjects at moderate, high, and very high risk for CKD were significantly higher (*p* < 0.001) in the population with MetS than with non-MetS (ORs 3.4, 5.4, and 5.9, respectively), in the population with morbid MetS than with premorbid MetS (ORs 2.0, 3.0, and 2.7, respectively), in the population with morbid MetS than with non-MetS (ORs 4.9, 8.8, and 9.2, respectively), and in the population with premorbid MetS than with non-MetS (ORs 2.5, 3.0, and 3.4, respectively) (Appendix A, Appendix A). The percentages of subjects at high and very high risk for CVD (23.2% and 56.1%, respectively) were significantly higher (*p* < 0.001) in the population with MetS than with non-MetS (ORs 2.8 and 9.5, respectively) (Appendix A, Appendix A). The percentages of subjects at moderate, high, and very high risk for CVD (36.2%, 40.5%, and 23.3%, respectively) were significantly higher (*p* < 0.001) in the population with premorbid MetS than with non-MetS (ORs 2.1, 6.4, and 2.3, respectively) (Appendix A, Appendix A).

## 4. Discussion

### 4.1. Prevalence Rates

The worldwide prevalence rates of MetS vary depending on the diagnostic criteria used and the characteristics of the population to which each definition applies [17]. Differences in prevalence rates according to the definitions used by scientific societies are also observed in our study, being slightly higher when less restrictive glycaemic or WC criteria are used. The pioneering studies that brought to light the high prevalence of MetS arose from the Third National Health and Nutrition Examination Survey (NHANES III) in the United States, conducted between 1988 and 1994 in adults 20 years of age and over [18]. This survey reported a MetS prevalence of 22.7% according to NCEP/ATP-III criteria [10]. The impact on the MetS by modifying the FPG ≥ 100 mg/dL criterion increased from 37.9% according to NCEP/ATP-III [10] to 43.5% according to the revised definition of AHA/NHLBI [11]. The NHANES 2003–2006 [19] reported that 34.0% of adults 20 years of age and over (35.1% in males; 32.6% in females) met MetS according to the 2009-JS criteria [13]. Other major studies such as the San Antonio Heart and Framingham Offspring Studies [20], Women’s Health Study [21], and West of Scotland Coronary Prevention Study (WOSCOPS) [22], in addition to showing similar prevalence rates, reported a nearly three-fold increased risk of CHD and an almost four-fold increased risk of developing DM among the MetS population. The MORGAN Project [23] assessed the MetS prevalence from 36 cohorts of European people between 19 and 78 years of age, being 14.2% according to IDF criteria [12] and 20.0% according to AHA/NHLBI criteria [11], with lower prevalence rates due to the younger age of the participants and because the presence of DM was considered as a glycaemic criterion in both cases. In Spain, the DARIOS study [24] published data from 11 cohorts of people aged 35 to 74 years. The prevalence rates of premorbid MetS and MetS according to the 2009-JS criteria [13] were 25% and 31%, respectively. The ENRICA study [25] published the MetS prevalence rates among people with a mean age of approximately 45.4 years. The prevalence of premorbid MetS and MetS were 16.9% and 22.7%, respectively. On the other hand, the di@bet.es study [26] reported MetS prevalence rates of 34.0% according to NCEP/ATP-III criteria [10], and 38.8% according to IDF criteria [12].

Our results showed that the MetS prevalence rates according to the 2009-JS criteria [13] increase with age and were slightly higher (36.0%) than those reported above, probably due to the older mean age of our study population. Premorbid MetS prevalence was also slightly lower (14.5%) and morbid MetS prevalence was slightly higher (21.5%), probably because the prevalence of DM and ASCVD was higher with increasing age. However, these data are still lower than the most recent data from NHANES, which showed a continuous increase in the MetS prevalence from 37.6% in 2011 to 41.8% in 2018 [27] due to significant increases in BMI, WC, CUN-BAE, DBP, FPG, and HbA1c [28].

### 4.2. Disorders and Clinical Conditions Other than Defining Criteria for MetS

Most clinical variables analysed showed worse results in the population with MetS than in the non-MetS population, probably due to their better health status and lower disease burden. The presence of MetS has been considered a consequence of the sedentary lifestyle of civilisation, easy access to food and exercise withdrawal, favouring an inflammatory state, and insulin resistance that can affect all the systems of the body with greater or lesser intensity, recently known as CKM syndrome [15,29]. A recent meta-analysis showed that long-time sedentary behaviour was associated with an increased risk of MetS regardless of the duration of physical activity, although no solid conclusion was drawn about the linearity of this association [30]. Our data analyses also suggest that both physical inactivity and alcoholism show a slight association with MetS. A recent review from the European Association for the Study of the Liver reported that MetS increases the risk of liver-related outcomes, regardless of the level of alcohol consumption, probably due to the complex relationships between alcohol consumption and MetS criteria [31]. Metabolic dysfunction-associated steatotic liver disease can be a cause but also a consequence of MetS [32]. A value FLI between 60 and 100 can be used as an accurate predictor of hepatic steatosis [33]. Our study also shows an independent association of high-FLI values with premorbid MetS.

The Cardiovascular Risk Survey conducted in China showed that WtHR in men and TG/HDL-c in women were the best predictors of MetS according to IDF criteria [12,34]. Other studies report that an elevated TG/HDL-c ratio may be independently associated with an increased risk of cardiovascular events [35] and that it is a risk marker for MetS and CVD [36]. A recent meta-analysis [37] showed that LAP was an accurate predictor for MetS in adults, better than other adiposity indicators such as BMI, WtHR, and WC. A NHANES analysis [38] showed that the TyG index had a high diagnostic value for MetS, even slightly outperforming the traditional homeostasis model assessment (HOMA). We found that many disorders other than the defining criteria for MetS were strongly associated with it. Overweight, obesity, high-AIP, and high-LAP index were associated with MetS, with overweight being the weakest. In addition, the high-TyG index was independently associated with MetS and with morbid MetS; high-VAI showed independent associations with MetS and premorbid MetS; the high-LAP index was only independently associated with premorbid MetS; and high-WtHR, EA, and hypercholesterolaemia showed independent associations with MetS (premorbid and morbid), with EA standing out with premorbid MetS.

Some lipid profile parameters such as TC and LDL-C showed lower concentrations in subjects with MetS due to the increased use of LLT, especially those with the morbid pattern. However, our analyses confirm that both hypercholesterolaemia and HTN were independently associated with MetS, more strongly with morbid MetS than with premorbid MetS. Furthermore, HF, AF, and erectile dysfunction were associated with premorbid MetS, and all CVDs (CHD, stroke, PAD, HF, and AF) and erectile dysfunction were strongly associated with MetS. Both CHD and CVDs as a whole showed an independent association with MetS. Likewise, HF and AF also showed an independent association with morbid MetS. Indeed, the percentages of subjects at high- and very high- cardiovascular risk was very high (79.3% [OR 14.0]), being 63.8% (OR 6.4) in patients with premorbid MetS. Some alterations observed for MetS patients in our study could be foreseen. However, other variables found are not strictly included in the different definitions of MetS and could be ignored in clinical practice. Reaven himself revised the meaning of MetS and stated that labelling a patient with MetS was not the ultimate goal but rather the search for other risk factors after identifying a major risk factor [39]. It is worth highlighting the factors associated with the presence of premorbid MetS in the multivariate analysis (HTN, hypercholesterolaemia, EA, high-VAI, high-LAP, FLI ≥ 60, WtHR ≥ 0.60) since they allow their identification to begin the search for other risk factors, improve lifestyles or start pharmacological treatment. Due to the high prevalence of EA (56.5%) and hypercholesterolemia (46.5%) found in patients without MetS, the presence of these disorders may alert for early detection of a large number of subjects prone to develop premorbid MetS in whom emphasis should be placed on improving lifestyles behaviours.

CKD and MetS are causal and influence each other [40]. Some meta-analyses [41,42] suggest that the MetS and its components are independently associated with a significant 50% increase in CKD risk and that the risk of developing incident CKD increased with a greater number of MetS components. CKM syndrome is based on EA (identified by being overweight, obesity, and abdominal obesity), dysfunctional adipose tissue (reflected by impaired glucose tolerance and hyperglycaemia), metabolic risk factors (hypertriglyceridaemia, HTN, MetS, and type 2 DM), and moderate or high risk of CKD [15]. CKM syndrome is a condition in which a purely metabolic alteration is responsible for heart and kidney diseases that coexist [43,44,45]. It is a more precise approximation than the MetS provides for assessing the overall risk of patients, not only for DM or ASCVD but also for CKD. However, a conceptual reappraisal of CKM syndrome is still needed [46]. It remains to be determined what other non-redundant clinical conditions are associated with MetS. For example, CKM syndrome includes, in addition to MetS itself, hyperglycaemia, abdominal obesity, hypertriglyceridaemia, and increased blood pressure, which are already included as MetS criteria. On the other hand, MetS focuses on metabolic factors, but it does not include EA and CKD risk. Our data analyses showed that hyperuricaemia, albuminuria, low eGFR, and CKD were strongly associated with Mets (premorbid and morbid) and that both albuminuria and CKD were independently associated with morbid MetS. Indeed, the percentage of subjects with MetS at moderate, high, or very high risk for CKD was very high (19.6% [OR 4.4]), being 13.3% (OR 2.8) in patients with premorbid MetS and 28.2% (OR 7.1) in patients with morbid MetS.

### 4.3. Limitations, Strengths, and Considerations

The main limitations of our study were the possible underestimation of prevalence rates because the institutionalised population was excluded from the study, the inability to preclude definitive conclusions about causal relationships or estimate incidence rates due to its observational design, variability between interviewers, and heterogeneity of laboratory equipment and measurement precision. The values of variables that were not reported in all study subjects were few, occurred at random, and were proportionally similar in the comparison groups, although this could imply minimal confounding in the comparative analysis between subjects with and without MetS. Results reporting associations between many factors or clinical conditions and the presence of MetS should be interpreted as speculative and with caution because multiple comparisons using ORs could increase the risk of obtaining a statistically significant association by chance (familywise error or alpha-inflation phenomenon). The higher intensity of GLT, BPLT, LLT, and ULT among people with MetS influences decreasing the values of the outcome variables and may be relevant, especially for the lipid profile parameters.

Key strengths include a large sample of people aged 18 to 102 years recruited using a population-based random method, and analysis of a broad spectrum of cardiovascular, renal, and metabolic variables other than those defining MetS. The results reported herein are biologically plausible and consistent with the available scientific information, update the adjusted prevalence rates of MetS-related variables in the overall population, and help to understand the relationship between the Mets and disorders other than the criteria that define it. MetS is a modifiable risk factor for DM and ASCVD, the dimension of which constitutes a serious health problem and a real challenge for clinical practice. The ongoing knowledge of multiple variables associated with MetS and their multidirectional interrelations requires research into factors that are independent of each other and that encompass a common and comprehensive disorder such as CKM syndrome. Regarding the clinical implications of our results, certain preventive strategies can be suggested that should be recommended to younger people, especially those with premorbid MetS. The age-related increase in the MetS prevalence shows a missed opportunity to improve lifestyle education at earlier ages since people with healthier lifestyle behaviours (physical activity, non-smoking, healthy diet, and BMI < 25 kg/m^2^) have a lower prevalence of MetS [47,48]. The high prevalence of MetS found in this study calls for more intensive healthcare and comprehensive approaches not only for its diagnostic criteria but also for other CKM disorders to reduce the DM incidence and minimise the long-term cardiovascular and renal risks. We hope that this study will help to update MetS prevalence and better understand the magnitude and importance of many CKM disorders associated with MetS other than its defining criteria.

## 5. Conclusions

Thirty-six percent of the overall adult population meets the criteria for the so-called harmonised MetS, 39.8% in men and 33.5% in women. More than one-fifth of adults meet premorbid MetS criteria, and almost one-sixth have MetS with DM and/or ASCVD. In addition to the criteria defining MetS, other medical conditions show an independent association with MetS, highlighting EA for premorbid MetS and HTN, hypercholesterolaemia, HF, and other CKM factors for morbid MetS. Almost two-fifths of people with MetS are at moderate, high or very high risk of CKD, and four-fifths are at high or very high cardiovascular risk. Based on this, we believe that it is necessary to broaden the concept of MetS and conduct further studies on which are the independent risk factors for DM, CVDs, and CKD in order to define a comprehensive disorder such as CKM syndrome.

## Figures and Tables

**Figure 1 biomedicines-13-00590-f001:**
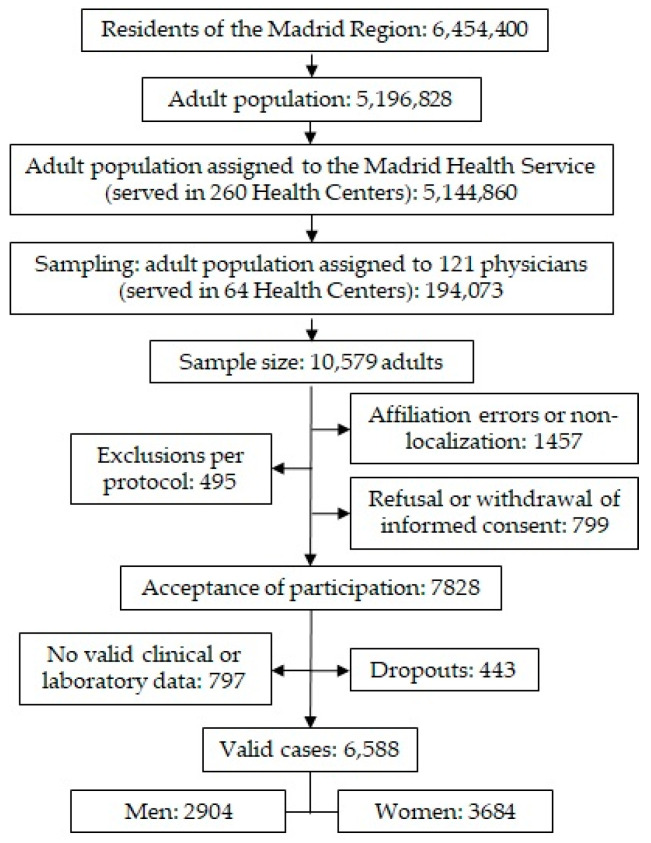
Flowchart for sampling and selection of study subjects.

**Figure 2 biomedicines-13-00590-f002:**
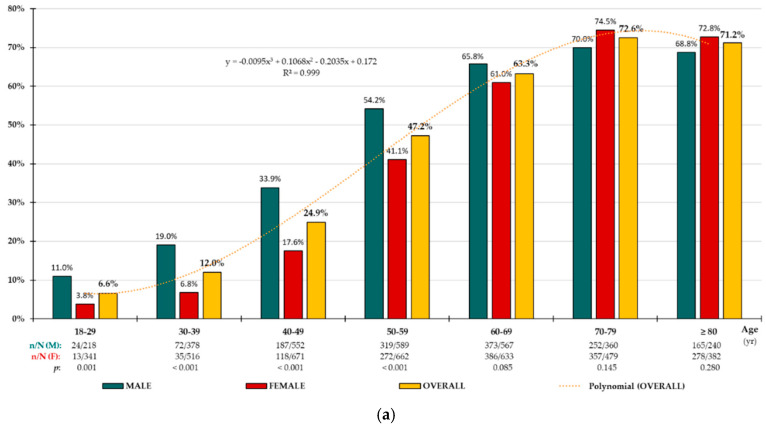
Age-specific prevalence rates of MetS (**a**), premorbid MetS (**b**), and morbid MetS (**c**). n: number of cases; N: sample size; M: male; F: female; *p*: *p*-value of the difference in percentages (M vs. F).

**Figure 3 biomedicines-13-00590-f003:**
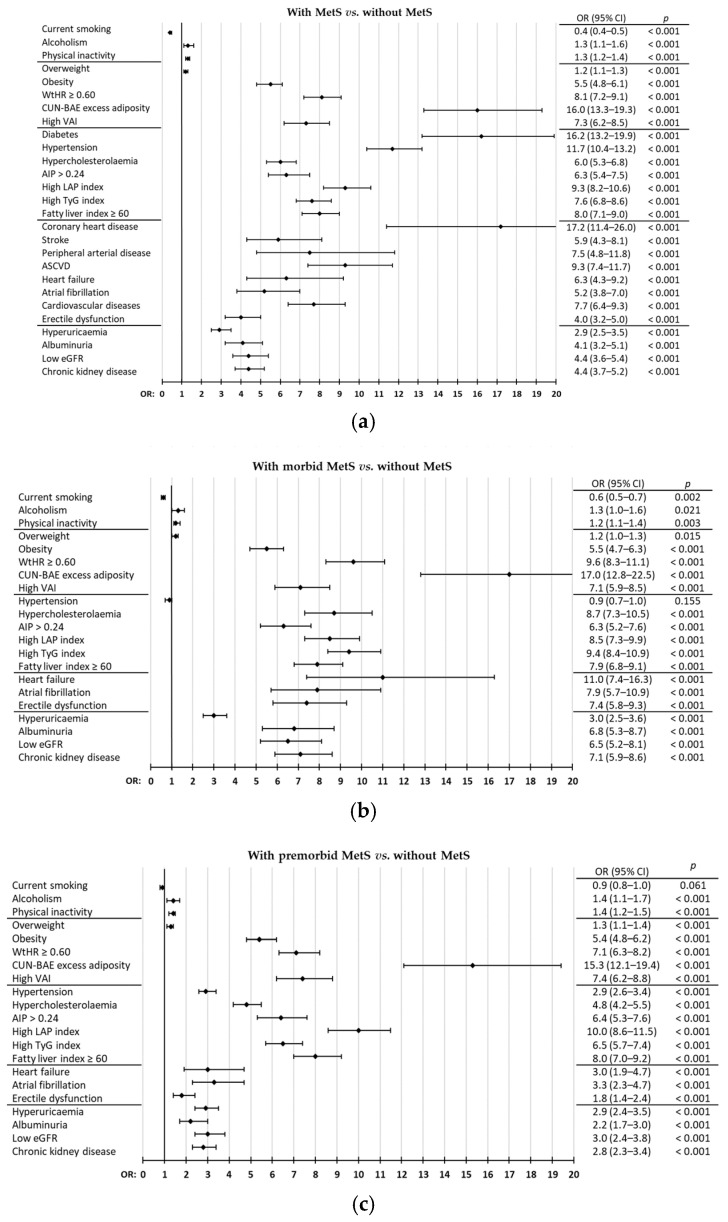
Diseases and clinical conditions in populations with and without MetS (**a**), with morbid MetS and without MetS (**b**), with premorbid MetS and without MetS (**c**). AIP: atherogenic index of plasma; ASCVD: atherosclerotic cardiovascular disease; CI: confidence interval; CUN-BAE: according to its acronym in Spanish, *Clínica Universitaria de Navarra*—Body Adiposity Estimator; eGFR: estimated glomerular filtration rate; LAP: lipid accumulation product index; OR: odds ratio; TyG: triglyceride-glucose index; VAI: visceral adiposity index; WtHR: waist-to-height ratio.

**Figure 4 biomedicines-13-00590-f004:**
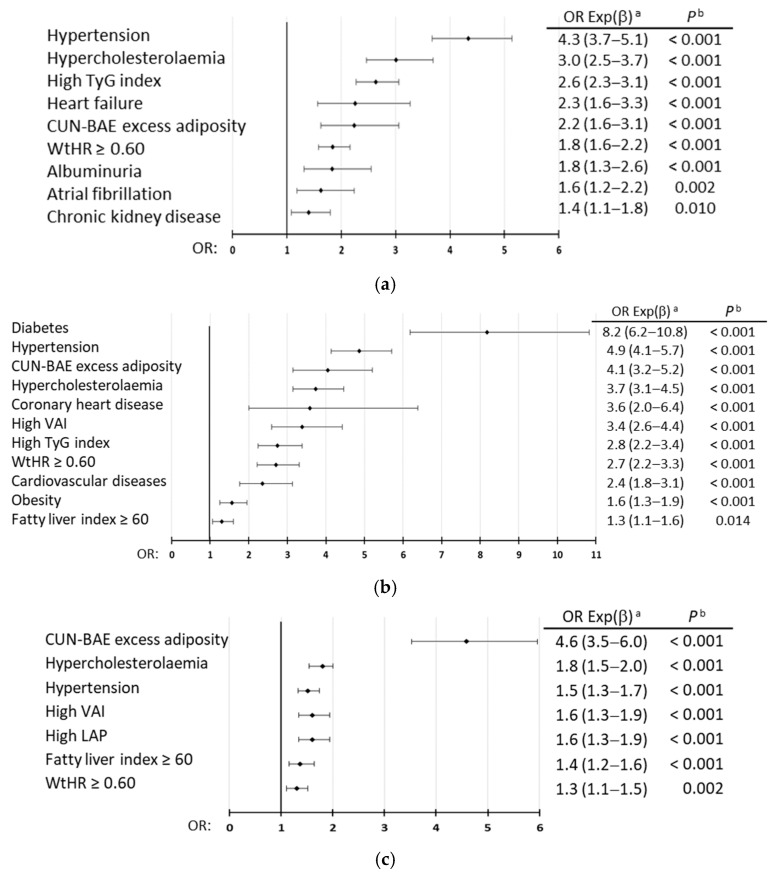
Multivariate analysis of diseases and medical conditions for MetS (**a**), for morbid MetS (**b**), and for premorbid MetS (**c**). ^a^ OR Exp (β): odds ratio (95% confidence interval); ^b^ *p*: *p*-value of Wald test with one degree of freedom. CUN-BAE: according to its acronym in Spanish, *Clínica Universitaria de Navarra*—Body Adiposity Estimator; LAP: lipid accumulation product index; TyG: triglyceride-glucose index; VAI: visceral adiposity index; WtHR: waist-to-height ratio.

**Table 1 biomedicines-13-00590-t001:** Percentages of criteria for the diagnosis of MetS.

	With MetS N = 2851 No. (%)	Without MetS N = 3737 No. (%)	OR (95% CI)	*p*	OR Exp (β) (95% CI)
WC ≥ 102 cm for men WC ≥ 88 cm for women	2036 (71.4)	886 (23.7)	8.0 (7.2–9.0)	<0.001	11.2 (9.3–13.6)
TG ≥ 150 mg/dL	1490 (52.3)	457 (12.2)	7.9 (7.0–8.9)	<0.001	13.1 (10.5–16.3)
HDL-c < 40 mg/dL for men HDL-c < 50 mg/dL for women	1289 (45.2)	530 (14.2)	5.0 (4.4–5.6)	<0.001	8.4 (6.8–10.5)
SBP ≥ 130 mmHg or DBP ≥ 85 mmHg or with BPLT in patients with HTN	2334 (81.6)	985 (26.4)	12.6 (11.2–14.2)	<0.001	19.8 (16.1–24.4)
FPG ≥ 100 mg/dL or with GLT	1970 (69.1)	515 (13.8)	14.0 (12.4–15.8)	<0.001	21.0 (17.1–25.8)

CI: confidence interval; OR: odds ratio; OR Exp (β): multivariate analysis odds ratio; *p*: *p*-value; BPLT: blood pressure-lowering drug therapy; DBP: diastolic blood pressure; FPG: fasting plasma glucose; GLT: glycaemic-lowering drug therapy; HDL-c: high-density lipoprotein cholesterol; HTN: arterial hypertension; SBP: systolic blood pressure; TG: triglycerides; WC: waist circumference.

**Table 2 biomedicines-13-00590-t002:** Prevalence rates of MetS according to 2009 Joint Statement, NCEP/ATP-III and IDF criteria.

	Crude Prevalence Rates				Age-Adjusted Prevalence Rates		
	Male % (95% CI)	Female % (95% CI)	*p*	Overall % (95% CI)	Male (%)	Female (%)	Overall (%)
Morbid 2009-JS	23.0 (21.5–24.6)	15.0 (13.9–16.2)	<0.001	18.6 (17.6–19.5)	17.2	12.3	14.5
Premorbid 2009-JS	24.9 (23.3–26.5)	24.6 (23.2–26.0)	0.777	24.7 (23.7–25.8)	22.6	21.2	21.5
2009-JS	47.9 (46.1–49.8)	39.6 (38.0–41.2)	<0.001	43.3 (42.1–44.5)	39.8	33.5	36.0
NCEP/ATP-III	44.1 (42.3–45.9)	36.5 (35.0–38.1)	<0.001	39.9 (38.7–41.0)	36.5	30.8	33.1
IDF	43.8 (42.0–45.6)	39.6 (38.0–41.2)	0.001	41.4 (40.2–42.6)	37.1	33.9	34.9

MetS: metabolic syndrome; CI: confidence interval; *p*: *p*-value of difference in percentages; 2009-JS: 2009 Joint Statement of the IDF, AHA/NHLBI, WHF, IAS, and IASO [13]; NCEP/ATP-III: National Cholesterol Education Program’s Adult Treatment Panel III report [10]; IDF: International Diabetes Federation [12].

**Table 3 biomedicines-13-00590-t003:** Clinical characteristics of populations with and without MetS.

	With MetS No. 2851	Without MetS No. 3737	With MetS vs. Without MetS	
	Mean (SD)	Mean (SD)	*p*	Cohen’s *d* (95% CI)
Age yr	64.5 (14.2)	48.0 (16.4)	<0.001	1.1 (1.0; 1.1)
BMI kg/m^2^	30.1 (4.9)	25.5 (4.4)	<0.001	1.0 (1.0; 1.0)
WC cm	101.2 (12.4)	87.3 (12.2)	<0.001	1.1 (1.1; 1.2)
WtHR	0.62 (0.08)	0.53 (0.07)	<0.001	1.2 (1.2; 1.3)
CUN-BAE adiposity	38.4 (7.9)	31.9 (8.1)	<0.001	0.8 (0.8; 0.9)
SBP mmHg	128.5 (14.5)	116.9 (14.3)	<0.001	0.8 (0.8; 0.9)
DBP mmHg	76.0 (9.5)	71.3 (9.5)	<0.001	0.5 (0.4; 0.5)
FPG mg/dL	107.6 (31.5)	87.2 (15.9)	<0.001	0.9 (0.8; 1.0)
HbA1c % #	6.03 (1.01)	5.26 (0.56)	<0.001	1.0 (0.9; 1.0)
TC mg/dL	192.1 (41.0)	193.3 (38.0)	0.257	0.1 (−0.1; 0.0)
HDL-c mg/dL	50.4 (14.0)	58.2 (14.3)	<0.001	−0.6 (−0.6; −0.5)
LDL-c mg/dL #	112.3 (35.6)	115.6 (33.6)	<0.001	−0.1 (−0.2; 0.0)
TG mg/dL	149.9 (94.2)	98.1 (65.3)	<0.001	0.7 (0.6; 0.7)
TG/HDL-c	3.39 (2.98)	1.87 (1.93)	<0.001	0.6 (0.6; 0.7)
AIP	0.07 (0.28)	−0.17 (0.25)	<0.001	0.9 (0.9; 1.0)
TyG index	8.83 (0.58)	8.23 (0.49)	<0.001	1.1 (1.1; 1.2)
LAP index	68.2 (47.3)	29.9 (23.9)	<0.001	1.1 (1.0; 1.2)
VAI	2.42 (2.01)	1.30 (1.19)	<0.001	0.7 (0.7; 0.8)
SUA mg/dL #	5.40 (1.50)	4.63 (1.38)	<0.001	0.5 (0.5; 0.6)
AST U/L #	23.6 (18.2)	22.6 (55.7)	0.422	0.0 (−0.1; 0.1)
ALT U/L #	27.0 (18.5)	23.2 (15.5)	<0.001	0.2 (0.2; 0.3)
GGT U/L #	41.5 (65.9)	27.3 (33.7)	<0.001	0.3 (0.2; 0.3)
FLI 0–100 #	64.4 (25.8)	30.3 (25.1)	<0.001	1.3 (1.3; 1.4)
Creatinine mg/dL	0.89 (0.33)	0.81 (0.26)	<0.001	0.3 (0.2; 0.3)
eGFR mL/min/1.73 m^2^	82.1 (19.7)	97.0 (18.7)	<0.001	−0.8 (−0.8; −0.7)
uACR mg/g	24.3 (82.7)	10.4 (33.8)	<0.001	0.2 (0.2; 0.3)

CI: confidence interval; Cohen’s *d*; effect size of standardised mean difference according to the proximity to the following absolute *d*-values: 0.2 small; 0.5 medium; 0.8 large; MetS: metabolic syndrome; *p*: *p*-value of the difference in means; SD: standard deviation. AIP: atherogenic index of plasma; ALT: alanine aminotransferase (# No. with MetS, without MetS: 2781, 3641, respectively); AST: aspartate aminotransferase (# No. with MetS, without MetS: 2161, 2660, respectively); BMI: body mass index; CUN-BAE-adiposity: adiposity or CUN-BAE body fat index; DBP: diastolic blood pressure; eGFR: estimated glomerular filtration rate; FPG: fasting plasma glucose; FLI: fatty liver index (# No. with MetS, without MetS: 2657, 3451, respectively); GGT: gamma-glutamyl transferase (# No. with MetS, without MetS: 2657, 3451, respectively); HbA1c: glycated haemoglobin A1c (# No. with MetS, without MetS: 2559, 2674, respectively); HDL-c: high-density lipoprotein cholesterol; LAP: lipid accumulation product index; LDL-c: low-density lipoprotein cholesterol (# No. with MetS, without MetS: 2803, 3723, respectively); SBP: systolic blood pressure; SUA: serum uric acid (# No. with MetS, without MetS: 2814, 3675, respectively); TC: total cholesterol; TG: triglyceride; TyG: triglyceride-glucose index; uACR: urine albumin-creatinine ratio; VAI: visceral adiposity index; WC: waist circumference; WtHR: waist-to-height ratio.

**Table 4 biomedicines-13-00590-t004:** Clinical characteristics of populations with morbid MetS, premorbid MetS, and without MetS.

	Morbid MetS No. 1222	Premorbid MetS No. 1629	Morbid MetS vs. Premorbid MetS		Morbid MetS vs. Without MetS		Premorbid MetS vs. Without MetS	
	Mean (SD)	Mean (SD)	*p*	Cohen’s *d* (95% CI)	*p*	Cohen’s *d* (95% CI)	*p*	Cohen’s *d* (95% CI)
Age yr	68.9 (12.6)	61.2 (14.5)	<0.001	0.6 (0.5; 0.6)	<0.001	1.3 (1.3; 1.4)	<0.001	0.8 (0.8; 0.9)
BMI kg/m^2^	30.3 (5.3)	30.0 (4.6)	0.146	0.1 (0.0; 0.1)	<0.001	1.0 (1.0; 1.1)	<0.001	1.0 (1.0; 1.1)
WC cm	102.0 (12.9)	100.7 (12.0)	0.006	0.1 (0.0; 0.2)	<0.001	1.2 (1.1; 1.3)	<0.001	1.1 (1.1; 1.2)
WtHR	0.63 (0.08)	0.62 (0.07)	<0.001	0.1 (0.1; 0.2)	<0.001	1.4 (1.3; 1.4)	<0.001	1.3 (1.2; 1.3)
CUN-BAE adiposity	38.3 (8.2)	38.6 (7.7)	0.317	0.0 (−0.1; 0.0)	<0.001	0.8 (0.7; 0.9)	<0.001	0.8 (0.8; 0.9)
SBP mmHg	128.9 (14.9)	128.1 (14.2)	0.145	0.1 (0.0; 0.1)	<0.001	0.8 (0.8; 0.9)	<0.001	0.8 (0.7; 0.8)
DBP mmHg	74.7 (9.6)	77.0 (9.4)	<0.001	−0.2 (−0.3; −0.2)	<0.001	0.4 (0.3; 0.4)	<0.001	0.6 (0.5; 0.7)
FPG mg/dL	124.5 (40.3)	94.9 (11.7)	<0.001	1.1 (1.0; 1.1)	<0.001	1.5 (1.5; 1.6)	<0.001	0.5 (0.5; 0.6)
HbA1c % #	6.60 (1.19)	5.55 (0.40)	<0.001	1.2 (1.1; 1.3)	<0.001	1.7 (1.6; 1.7)	<0.001	0.6 (0.5; 0.6)
TC mg/dL	176.5 (39.0)	203.9 (38.6)	<0.001	−0.7 (−0.8; −0.6)	<0.001	−0.4 (−0.5; −0.4)	<0.001	0.3 (0.2; 0.3)
HDL-c mg/dL	49.6 (13.8)	51.0 (14.1)	0.181	−0.1 (−0.1; 0.0)	<0.001	−0.6 (−0.7; −0.5)	<0.001	−0.5 (−0.6; −0.5)
LDL-c mg/dL #	98.2 (33.3)	122.9 (33.6)	<0.001	−0.7 (−0.8; −0.7)	<0.001	−0.5 (−0.6; −0.5)	<0.001	0.2 (0.2; 0.3)
TG mg/dL	145.0 (81.3)	153.6 (102.8)	0.016	−0.1 (−0.2; 0.0)	<0.001	0.7 (0.6; 0.7)	<0.001	0.7 (0.6; 0.8)
TG/HDL-c	3.32 (2.55)	3.44 (3.27)	0.288	0.0 (−0.1; 0.0)	<0.001	0.6 (0.5; 0.6)	<0.001	0.5 (0.4; 0.5)
AIP	0.07 (0.28)	0.08 (0.28)	0.340	0.0 (−0.1; 0.0)	<0.001	0.9 (0.9; 1.0)	<0.001	1.0 (0.9; 1.0)
TyG index	8.94 (0.63)	8.75 (0.51)	<0.001	0.3 (0.3; 0.4)	<0.001	1.3 (1.3; 1.4)	<0.001	1.0 (1.0; 1.1)
LAP index	67.4 (46.7)	68.8 (47.8)	0.435	0.0 (−0.1; 0.0)	<0.001	1.2 (1.1; 1.3)	<0.001	1.2 (1.1; 1.2)
VAI	2.36 (1.78)	2.47 (2.16)	0.148	0.0 (−0.1; 0.0)	<0.001	0.8 (0.7; 0.8)	<0.001	0.8 (0.7; 0.8)
SUA mg/dL #	5.43 (1.54)	5.38 (1.46)	0.436	0.0 (0.0; 0.1)	<0.001	0.6 (0.5; 0.6)	<0.001	0.5 (0.5; 0.6)
AST U/L #	24.0 (24.8)	23.4 (11.2)	0.476	0.0 (0.0; 0.1)	0.146	0.0 (−0.1; 0.1)	0.195	0.0 (−0.1; 0.1)
ALT U/L #	27.4 (21.5)	26.7 (15.8)	0.345	0.0 (0.0; 0.1)	<0.001	0.2 (0.2; 0.3)	<0.001	0.2 (0.2; 0.3)
GGT U/L #	43.1 (54.6)	40.2 (73.2)	0.261	0.0 (0.0; 0.1)	<0.001	0.4 (0.3; 0.5)	<0.001	0.3 (0.2; 0.3)
FLI 0–100 #	64.6 (26.4)	64.2 (25.4)	0.683	0.0 (−0.1; 0.1)	<0.001	1.3 (1.3; 1.4)	<0.001	1.3 (1.3; 1.4)
Creatinine mg/dL	0.92 (0.38)	0.86 (0.28)	<0.001	0.2 (0.1; 0.3)	<0.001	0.4 (0.3; 0.4)	<0.001	0.2 (0.1; 0.3)
eGFR mL/min/1.73 m^2^	78.1 (20.3)	85.0 (18.8)	<0.001	−0.4 (−0.4; −0.3)	<0.001	−1.0 (−1.1; −0.9)	<0.001	−0.6 (−0.7; −0.6)
uACR mg/g	37.2 (114.3)	14.6 (44.2)	<0.001	0.3 (0.2; 0.4)	<0.001	0.4 (0.4; 0.5)	<0.001	0.1 (0.0; 0.2)

Abbreviations with the same meaning as those indicated in the footnotes of Table 3. ALT #: No. morbid, premorbid, without MetS: 1195, 1586, 3641, respectively; AST #: No. morbid, premorbid, without MetS: 915, 1246, 2660, respectively; FLI #: No. morbid, premorbid, without MetS: 1140, 1517, 3451, respectively; GGT #: No. morbid, premorbid, without MetS: 1140, 1517, 3451, respectively; HbA1c #: No. morbid, premorbid, without MetS: 1184, 1375, 2674, respectively; LDL-c #: No. morbid, premorbid, without MetS: 1203, 1600, 3723, respectively; SUA #: No. morbid, premorbid, without MetS: 1201, 1613, 3675, respectively.

## Data Availability

The original contributions presented in this study are included in the article/Appendix A.

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
