# Peer review of "From Metabolic Syndrome to Cardio-Kidney-Metabolic Syndrome in the SIMETAP Study: Prevalence Rates of Metabolic Syndrome and Its Independent Associations with Cardio-Renal-Metabolic Disorders Other than Its Defining Criteria"

_biomedicines, 2025, doi:10.3390/biomedicines13030590_

Round 1

Reviewer 1 Report

Comments and Suggestions for Authors

The aim of presented study was to establish the prevalence of metabolic syndrome (MS) in cross-sectional observatory multi-centre study conducted in Spain.  Four different definitions of MS were used . The authors analyzed pre-morbid (MS without DM and ASCVD)  and morbid sub-populations.

Metabolic syndrome  and its prevalence were analyzed in different age sub-groups . Many demographic and laboratory variables were subjected to thorough statistical analysis. Very detailed descriptions under the tables  make the analysis of the data much more difficult in my opinion. I have no objections to the statistical part. Conclusions consistent with the evidence and  presented arguments.

There is a lot of inconsistency in text. In some parts number are used , in others you put numbers with words . Example : line  139-146; 525-525   etc.

In tables descriptions are too long.

In my opinion there is no need to explain abbreviations in every line where they are used.

Author Response

1. Summary

Thank you very much for taking the time to review this manuscript. Please find the detailed responses below and the revisions/corrections highlighted in red on the manuscripts in the re-submitted files.

2. Point-by-point response to Comments and Suggestions for Authors

Comments 1: Metabolic syndrome and its prevalence were analyzed in different age sub-groups. Many demographic and laboratory variables were subjected to thorough statistical analysis. Very detailed descriptions under the tables make the analysis of the data much more difficult in my opinion. I have no objections to the statistical part. Conclusions consistent with the evidence and presented arguments.

In tables descriptions are too long.

In my opinion there is no need to explain abbreviations in every line where they are used.

Response 1: We agree with these comments. Therefore, we have replaced the previous versions of Tables 3 and 4 to the new versions, in which the variables eAG, residual cholesterol, non-HDL and non-HDL-c/HDL-c have been removed, and the errors that existed for the variables TG/HDL-C and AIP have been corrected. In addition, following your suggestions, we have also considerably reduced the footnotes in all Tables and Figures.

You can find these changes in Tables 1, 2, 3, and 4, and Figures 2, 3, and 4 of the revised manuscript.

Comments 2: There is a lot of inconsistency in text. In some parts number are used, in others you put numbers with words. Example: line 139-146; 525-525   etc.

Response 2: Thank you very much for these considerations. Since it is not possible to start a sentence by writing numerical digits but rather by using words, we have decided to change the beginning of sentences with words and express numerical data with digits inside the sentence, in order to improve the consistency in text and the uniformity of writing.

These changes can be found in page number 4, second paragraph, line from 147 to 149; page number 9, second paragraph, line from 257 to 259; and page number 10, first paragraph, line from 281 to 282 of the revised manuscript.

Reviewer 2 Report

Comments and Suggestions for Authors

The manuscript evaluates prevalence of metabolic syndrome and its associations with other cardio-renal-metabolic disorders among the participants of the SIMETAP study. The study included 6588 adult participants from Madrid, so the results provide an update of the prevalence rate of metabolic syndrome, both premorbid and morbid, in this area. The manuscript is very well written and the results are interesting and important. The use of electronic health records supports the relevance of the obtained results. Although Supplementary files are quite informing, I would suggest placing the Flowchart in the main text, as well as inclusion of the definition used for metabolic syndrome. Also, the authors should consider including a brief discussion on clinical implications of their results, in terms of preventive strategies that should be advised to premorbid metabolic syndrome patients.

Author Response

1. Summary

Thank you very much for taking the time to review this manuscript. Please find the detailed responses below and the revisions/corrections highlighted in red on the manuscripts in the re-submitted files.

2. Point-by-point response to Comments and Suggestions for Authors

Comments 1: The manuscript evaluates prevalence of metabolic syndrome and its associations with other cardio-renal-metabolic disorders among the participants of the SIMETAP study. The study included 6588 adult participants from Madrid, so the results provide an update of the prevalence rate of metabolic syndrome, both premorbid and morbid, in this area. The manuscript is very well written and the results are interesting and important. The use of electronic health records supports the relevance of the obtained results. Although Supplementary files are quite informing, I would suggest placing the Flowchart in the main text, as well as inclusion of the definition used for metabolic syndrome.

Response 1: Thank you very much for your considerations. We agree with these comments. We have placed the flowchart for sampling and selection of study subjects in the main text. You can find this change in Figure 1 of the revised manuscript.

We have included the definition used for metabolic syndrome in the main text. You can find this change in page number 3, second paragraph, line from 112 to 119 of the revised manuscript.

Comments 2: Also, the authors should consider including a brief discussion on clinical implications of their results, in terms of preventive strategies that should be advised to premorbid metabolic syndrome patients.

Response 2: Thank you for pointing this out. Accordingly, we have emphasized this issue in the discussion section. You can find this change in page number 14, third paragraph, line from 441 to 450 of the revised manuscript.

Reviewer 3 Report

Comments and Suggestions for Authors

Despite the effort made by the authors in collecting a huge sample and conducting a robust statistical analysis of the data, the results are easily predictable since we are comparing variables directly or indirectly attributable to the diagnostic criteria of the metabolic syndrome in a sample with MetS (both in subclinical and manifest form) vs. one without.

Author Response

1. Summary

Thank you very much for taking the time to review this manuscript. Please find the detailed responses below and the revisions/corrections highlighted in red on the manuscripts in the re-submitted files.

2. Point-by-point response to Comments and Suggestions for Authors

Comments 1: Despite the effort made by the authors in collecting a huge sample and conducting a robust statistical analysis of the data, the results are easily predictable since we are comparing variables directly or indirectly attributable to the diagnostic criteria of the metabolic syndrome in a sample with MetS (both in subclinical and manifest form) vs. one without.

Response 1: Thank you for pointing this out. We agree with the reviewer's comment that many of the alterations observed in the MetS patients in our study could be foreseen. However, other variables found are not strictly included in the different definitions of MetS and could be ignored in clinical practice. Reaven himself proposed that the MetS diagnosis was not the definitive objective, but that the identification of a major risk factor should prompt the search for other risk factors or the presence of subclinical disease that could evolve if corrective measures were not implemented. Of particular importance are the factors associated with the presence of premorbid MetS in the multivariate analysis (CUN-BAE excess adiposity, hypercholesterolaemia, hypertension, high-VAI, high-LAP, fatty liver index ≥60, WtHR ≥ 0.60), as they allow their identification to initiate the search for other risk factors, to improve lifestyles or to initiate pharmacological treatment when indicated according to clinical practice guidelines. Due to the prevalence found in patients without MetS in the SIMETAP study, the measurement of CUN-BAE excess adiposity present in 56.5% of participants without MetS and hypercholesterolaemia in 46.5% would be the variables that lead to a greater number of subjects prone to the development of MetS. The ease and simplicity of these tests makes them very useful tools in office daily practice for the detection of subjects prone to developing premorbid MetS in whom special emphasis must be placed on improving lifestyles.

You can find this change in page number 13, third paragraph, line from 393 to 404 of the revised manuscript.

Round 2

Reviewer 3 Report

Comments and Suggestions for Authors

The changes made to the text do not change the previous consideration. Despite the effort made by the authors to collect a huge sample and conduct a robust statistical analysis of the data, the results are easily predictable since we are comparing variables directly or indirectly attributable to the diagnostic criteria of the metabolic syndrome in a sample with MetS (both subclinical and manifest) compared to one without. Even the data related to unconventional markers (such as the TyG index or WtHR) are not surprising because they consider parameters used to define the presence of MetS.